# Predicting online participation through Bayesian network analysis

**Elizaveta Kopacheva**  *

Department of Political Science & Centre for Data Intensive Sciences and Applications (DISA), Linnaeus University, Växjö, Sweden

* elizaveta.kopacheva@lnu.se

## Abstract

Despite the fact that preconditions of political participation were thoroughly examined before, there is still not enough understanding of which factors directly affect political participation and which factors correlate with participation due to common background variables. This article scrutinises the causal relations between the variables associated with participation in online activism and introduces a three-step approach in learning a reliable structure of the participation preconditions' network to predict political participation. Using Bayesian network analysis and structural equation modeling to stabilise the structure of the causal relations, the analysis showed that only age, political interest, internal political efficacy and no other factors, highlighted by the previous political participation research, have direct effects on participation in online activism. Moreover, the direct effect of political interest is mediated by the indirect effects of internal political efficacy and age via political interest. After fitting the parameters of the Bayesian network dependent on the received structure, it became evident that given prior knowledge of the explanatory factors that proved to be most important in terms of direct effects, the predictive performance of the model increases significantly. Despite this fact, there is still uncertainty when it comes to predicting online participation. This result suggests that there remains a lot to be done in participation research when it comes to identifying and distinguishing factors that stimulate new types of political activities.

**Data Availability Statement:** All relevant data are within the manuscript and its Supporting information files.

**Funding:** The authors received no specific funding for this work.

## Introduction

With the development of Internet technologies, new forms of political activities, e.g., online activism, become more popular and gain potential to replace traditional political participation types, e.g., contacting politicians, signing petitions, etc. [1, 2]. It has been argued that online participation attracts new groups of people into political action as social networking systems (SNSs) provide access to resources [3], expedite engendering of identities [4, 5] and expose users to recruitment [6]. In the meanwhile, political participation is an essential element of the system of checks and balances in democratic societies [7–9]. Thus, it is essential to understand what motivates people to participate in new forms of political activities to facilitate the mobilisation of people into political action.

**Competing interests:** The authors have declared
that no competing interests exist.

Despite the fact that scholars started to investigate preconditions of political participation several decades ago [e.g., 10, 11], there is still not enough understanding of which factors directly affect political participation and which factors correlate with the variable. The Civic Voluntarism Model (CVM), which is well-known in political participation research, proposes that political motivations, i.e., political information, interest, efficacy and party identification, resources and recruitment are the main factors to determine political participation [12]. Over the years, the list of mobilising factors extended radically. Political participation scholars highlight such characteristics as political [13, 14] and social trust [15–17], placement on the left-right scale [18], external [19] and internal political efficacy [20, 21] as factors associating with political participation. However, only a limited number of studies examined causality between those factors (often, within small-scale experimental studies [e.g., 22, 23]). Thus, e.g., [24] found that lower levels of political trust are positively associated with non-institutionalised political participation (i.e., activities that aim to influence political decision-making indirectly, e.g., petition signing, boycotting, protesting [25, p. 188]) but did not identify if participation is affected by political trust. Moreover, it has not been examined before if knowing some characteristics of a person, e.g., if a person has a high political interest or low political trust, can help to predict the probability of a person to participate in one or another political action.

The reasons for the lack of such explications are the limitations of regression analysis as a method often used in political participation research [e.g., 12, 25]. When conducting regression analysis of the preconditions of political participation, researchers are forced to make an assumption that such characteristics as political interest, political efficacy, social trust, etc. are independent of each other, which is not necessarily the case.

Another type of analysis used in political participation research is structural equation modeling (e.g., applied by [21]) that allows analysing structural relationships, which may be interpreted as causal relations between the variables. Structural equation modeling estimates the interrelated multiple dependencies given exogenously and does not provide tools to infer the relationships from the data, which in the case of preconditions of political participation, is essential as there is not enough evidence to suggest causal relations between factors associated with participation.

Indeed, for a long time, experimental studies with interventions were considered the only type of analysis that allows inferring causal relations in a social context. Judea Pearl was one of the first researchers who proposed using Bayesian networks to infer causality [26]. Later, reasoning about causal relations based on the results of Bayesian network analysis was frequently discussed in the literature [e.g., 27–29]. It has been proposed that this method of research has the capacity to learn reliable structures of causal relations [e.g., see 30]. Moreover, Bayesian network analysis does not have limitations discussed in relation to the regression analysis (i.e., an assumption of independence between the variables), and decreases the number of assumptions to a minimum [31]. Overall, by being more flexible, Bayesian network analysis allows researchers to lessen the expenses related to experimental research and provides the ability to infer causality and predict events using available data, which is often collected within other studies and with other purposes. Hence, Bayesian network analysis is becoming more popular to infer causal relations even when examining complex social phenomena [e.g., 32, 33].

In this study, Bayesian network analysis is utilised in order to analyse preconditions of political participation and acquire a probability distribution table of online activism (i.e., citizens' activities that aim to "raise awareness about political issues" and mobilise citizens to participate in other more traditional forms of political participation, e.g., petition signing, protesting, contacting politicians, to promote political or institutional reforms [34]), as an example of political participation. Relying on the European Social Survey data [35] and the theoretical suggestions of the earlier research [12, 14, 21, 36] to limit the number of

explanatory variables (see S1 File for more details), this research is a unique analysis of the online activism precondition structure that gives not only a deeper understanding of what motivates people to participate online but also a capacity to predict political participation having limited prior knowledge about a person, e.g., person's social or political trust, income, level of education, age, etc.

The aims of this research are to answer the theoretical questions discussed before and more importantly to propose a methodological solution to the problem of inferring a reliable causal relation structure using available, however, restrictive survey data (in comparison to the data collected within experiments or specifically for the purposes of the study). In particular, this paper presents an innovative three-step approach of utilising the tools of Bayesian network analysis and structural equation modeling to acquire a reliable structure of causal relations between characteristics operationalised by survey questionnaires. This research emphasises the importance of using structural equation modeling once the causal structure has been learned with the tools of Bayesian network analysis. Structural equation modeling allows stabilising the variability in results when acquiring the causal structure using different Bayesian network algorithms, i.e., constrained-based, score-based and hybrid. Comparing received models applying structural equation modeling tools, it becomes possible to acquire a structure more reliable for further interference.

When used in the context of preconditions of online participation, the three-step analysis, presented in the paper, allowed to conclude that only age, political interest and internal political efficacy, i.e., the belief of an individual that one can influence political decision-making [37], have direct effects on online activism. Moreover, the direct effect of political interest is mediated by the effects of age and internal political efficacy, which affect political participation directly and indirectly via political interest. Contrary to the previous suggestions [19, 38–42], political and social trust as well as external political efficacy, which is often understood as the responsiveness of the political system to the political actions of citizens [37], are independent of online political participation given internal political efficacy. The results also suggest that offline mobilisation (via non-governmental organisations, trade unions, work places) is independent of online political participation given internal political efficacy and political interest. This result is consistent with the suggestions of [3], who proposed that offline mobilisation is associated only with offline political participation.

## Methods

Within the study, 30 ESS questions [35] that operationalise possible preconditions of political participation were analysed. The list of the variables includes such factors as party identification, political interest, internal and external political efficacy, highlighted in the state-of-the-art work on political participation preconditions of Verba, Schlozman and Brady [12]. In addition to those factors, the influence of other characteristics that are expected to be associated with political participation is examined. Thus, earlier, scholars suggested political trust to be negatively associated with participation in non-institutionalised activities [25, 38–41, 43] and social trust to be positively correlated with this type of participation [15, 44, 45].

Moreover, Verba, Schlozman and Brady [12] highlight the importance of resources and recruitment for any type of political participation suggesting that access to resources and mobilisation play a key role in stimulating political participation. Previous research also showed gender, nationality, income and educational level to be associated with political participation [12, 46], thus, those variables were also used for the analysis.

The dependent variable, online political participation, measured within the 2018 European Social Survey by the survey question "Have you. . . posted or shared anything about politics

online in the last 12 months" [35] and originally has a binary scale, where 1 represent the occurrence of online participation and 2—no occurrence. Table 1 shows how the rest of the variables were operationalised.

Exploratory factor analysis was applied to 12 out of 30 ESS questions to reduce the number of variables and operationalise social trust, political trust, external and internal political efficacy (see Table 2 for the factor loadings).

In order to perform Bayesian structure learning and acquire a reliable structure, all the variables, including those received as the result of the exploratory factor analysis, were discretised as suggested by [47] (see S1 File to know how the variables were originally measured and how they were transformed for the analysis). Despite the fact that discretisation caused some information loss, the step was necessary in order to follow conditional Gaussian distribution assumptions that apply when working with the mixed data, i.e., continuous and discrete variables. The conditional Gaussian distribution suggests that discrete nodes cannot have continuous parents [47, p. 128]. Following this assumption, we would be forced to presume that age cannot directly influence online political participation, which would affect the results.

Another constrain of the Bayesian network structure learning as a method to analyse factors influencing online activism is associated with the nature of a Bayesian network (BN) as a directed acyclic graph (DAG). Due to the fact that the structure cannot contain cycles, Bayesian network analysis does not allow children to affect parents, thus, simplification takes place. It is worth mentioning that no other methods allow directed graphs to be cyclic and cycles are possible only in undirected graphs. Hence, when evaluating the results, it is important to understand that only strong links between the variables are retained as Bayesian network algorithms aim to increase the ability to predict online participation.

When discussing the constraints of Bayesian network analysis to infer causal relations between the variables, the idea of causality must be also touched upon. Indeed, once interpreting edges as causal relations, one must be aware of the fact that only the relations between the variables as opposed to the relations between singular events are distinguished. Thus, Bayesian network analysis infers generic causality rather than the single-case one [48]. In other words, this article presents the dependencies between the variables instead of the causality in traditional understanding as event A leading to the occurrence of B [26].

Moreover, as for any other statistical inference method, the accuracy of Bayesian network structure learning is highly affected by missing or imprecise observations. Thus, estimating the performance of Bayesian network structure learning algorithms on several synthetic networks, [49] found that the accuracy of the structure learning decreases by 13%-28% if data contain from 5% to 10% missing values and by 18%-28% if there are 5%-10% inaccurate values. Thus, if both types of imprecise observations characterise the data, the decrease in accuracy ranges between 26% and 30% [49, p. 22]. For another thing, the same study showed that Bayesian network structure learning performs best on the datasets containing 100 000—1 000 000 observations [49, p. 15], which is often hard to reach when working with survey data.

The minimum number of observations needed for a correct Bayesian network structure learning is an ongoing topic of research. Earlier, it was proposed that this number depends on both, the total of nodes in the examined network and the complexity of the data [50], i.e., the number of categories within each variable and the number of missing and inaccurate values. In that regard, within this study, several robustness tests were performed to distinguish arcs that are not learned correctly (see S1 File).

A three-step analysis was conducted in order to learn a reliable structure of the network and acquire a set of conditional probability tables. The step-wise description of the procedure is presented in Fig 1.

**Table 1. Variables used for the analysis.**

| Variable name | Meaning | Operationalisation of | Values |
|---|---|---|---|
| pstplonl | Posted or shared anything about politics online in the last 12 months | Online activism | 1—Posted; 2—Did not post |
| sgnptit | Signed a petition in the last 12 months | Signing petition | 1—Signed; 2—Did not sign |
| contplt | Contacted a politician or government official during the last 12 months | Contacting politicians | 1—Contacted; 2—Did not contact |
| vote | Voted in the last national election | Participation in voting | 1–Voted; 2—Did not vote; 3—Not eligible |
| ppltrst | Most people can be trusted or you can't be too careful | Social trust | From 0 ("You can't be too careful") to 10 ("Most people can be trusted") |
| pplfair | Most people try to take advantage of you, or try to be fair | Social trust | From 0 ("Most people try to take advantage") to 10 ("Most people try to be fair") |
| pplhlp | Most of the time people helpful or mostly looking out for themselves | Social trust | From 0 ("mostly look out for themselves") to 10 ("mostly try to be helpful") |
| trstlgl | Trust in the legal system | Political trust | From 0 ("No trust at all") to 10 ("Complete trust") |
| trstplc | Trust in the police | Political trust | From 0 ("No trust at all") to 10 ("Complete trust") |
| trstplt | Trust in politicians | Political trust | From 0 ("No trust at all") to 10 ("Complete trust") |
| trstprt | Trust in political parties | Political trust | From 0 ("No trust at all") to 10 ("Complete trust") |
| trstprl | Trust in country's parliament | Political trust | From 0 ("No trust at all") to 10 ("Complete trust") |
| polintr | How interested in politics | Political interest | 1—"Very interested"; 2—"Quite interested"; 3—"Hardly interested"; 4—"Not at all interested" |
| psppsgva | Political system allows people to have a say in what government does | External political efficacy | 1—"Not at all"; 2—"Very little"; 3—"Some"; 4—"A lot"; 5—"A great deal" |
| psppipla | Political system allows people to have influence on politics | External political efficacy | 1—"Not at all"; 2—"Very little"; 3—"Some"; 4—"A lot"; 5—"A great deal" |
| clsprty | Is there a particular political party you feel closer to than all the other parties? | Party identification | 1—"Yes"; 2- "No" |
| lrscale | Placement on left right scale | Placement on the left-right scale | From 0 (left) to 10 (right) |
| actrolga | Able to take active role in political group | Internal political efficacy | 1—"Not at all able"; 2—"A little able"; 3—"Quite able"; 4—"Very able"; 5—"Completely able" |
| cptppola | Confident in own ability to participate in politics | Internal political efficacy | 1—"Not at all confident"; 2—"A little confident"; 3—"Quite confident"; 4—"Very confident"; 5—"Completely confident" |
| pdwrk | Doing last 7 days: paid work | Recruitment | 0—"Not marked"; 1—"Marked" |
| wrkorg | Worked in another organisation or association last 12 months | Recruitment | 1—"Yes"; 2—"No" |
| rlgblg | Belonging to particular religion or denomination | Recruitment | 1—"Yes"; 2—"No" |
| dscrgrp | Member of a group discriminated against in this country | Recruitment | 1—"Yes"; 2—"No" |
| mbtru | Member of trade union or similar organisation | Recruitment | 1—"Yes, currently"; 2—"Yes, previously"; 3—"No" |
| eduyrs | Years of full-time education completed | Education | Number of years |
| hincfel | Feeling about household's income nowadays | Income | 1—"Living comfortably on present income"; 2—"Coping on present income"; 3—"Difficult on present income"; 4—"Very difficult on present income" |
| brncntr | Were you born in country? | Nationality | 1—"Yes"; 2—"No" |
| gndr | Gender | Gender | 1—"Male"; 2—"Female" |
| agea | Age of respondent, calculated | Age | Number of years |
| cntry | Country | Country of origin | Country |

*Source*: ESS 2018 [35]. N = 36 015 individuals in 19 countries.

**Table 2. Operationalisation of social trust, political trust, external and internal political efficacy.**

| Operationalisation of | Variables | Factor loadings | Proportion of variance explained by the factor |
|---|---|---|---|
| Social trust | ppltrst | 0.751 | 0.57 |
| | pplfair | 0.802 | |
| | pplhlp | 0.711 | |
| Political trust | trstlgl | 0.764 | 0.64 |
| | trstplc | 0.646 | |
| | trstplt | 0.875 | |
| | trstprt | 0.847 | |
| | trstprl | 0.842 | |
| External political efficacy | psppsgva | 0.751 | 0.56 |
| | psppipla | 0.751 | |
| Internal political efficacy | actrolga | 0.757 | |
| | cptppola | 0.757 | 0.57 |

*Source*: ESS 2018 [35]. N = 36 015 individuals in 19 countries.

## Step 1: Bayesian network structure learning

Firstly, a set of the BN structure learning algorithms was applied to the network. All structure learning algorithms aim to find such model that maximises

$$Pr(G|D) \propto Pr(G)Pr(D|G) = Pr(G) \int Pr(D|G, \Theta)Pr(\Theta|G)d\Theta,$$

where $D$ is a dataset, $B = (G, \Theta)$ is a Bayesian network, in which the parameters of the global distribution of a set of variables $X = X_1, X_2, \ldots, X_p$ is denoted as $\Theta$ and G is a directed acyclic graph, $Pr(G|D)$ is the posterior probability of the DAG, $Pr(G)$ is the product of the prior distribution over the possible DAGs and $Pr(D|G)$ is the probability of the data [47]. All structure learning algorithms approach the task of formula maximisation differently. Score-based algorithms apply heuristic optimisation techniques assigning to each structure candidate a network score (i.e., Bayesian Information criterion (BIC) [51] or Bayesian Dirichlet equivalent (BDE) uniform [52] scores), which shows the model's goodness of fit, and trying to maximise it. Constraint-based algorithms use conditional independence tests to firstly, determine which pairs of variables are connected by an arc and cannot be d-separated; secondly, find v-structures and then, identify compelled arcs and their orientation. Finally, hybrid algorithms combine the approaches of the constraint-based and score-based algorithms trying to maximise the network score while restricting the results keeping only those structures, which satisfy certain conditions [47].

In this analysis, the performance of the following algorithms was tested: constraint-based Grow-Shrink [53], Incremental Association [54] and Max-Min Parents and Children [55]; score-based hill-climbing [47] and Tabu search [56] and hybrid Max-Min Hill Climbing [57] and Hybrid HPC [58]. Score-based and hybrid algorithms were successful in finding directed acyclic graph (DAG) structures. In the meanwhile, constraint-based algorithms found only partially directed structures, which is in line with the results previously reported by [49]. Earlier, it was suggested that score-based algorithms are "superior" to constraint-based algorithms when working with the data characterised by high volumes of noise [49, p. 24]. Thus, both TABU and hill-climbing (HC) showed high accuracy in learning the structures on the datasets including missing and inaccurate values [49]. Due to the fact that constraint-based algorithms

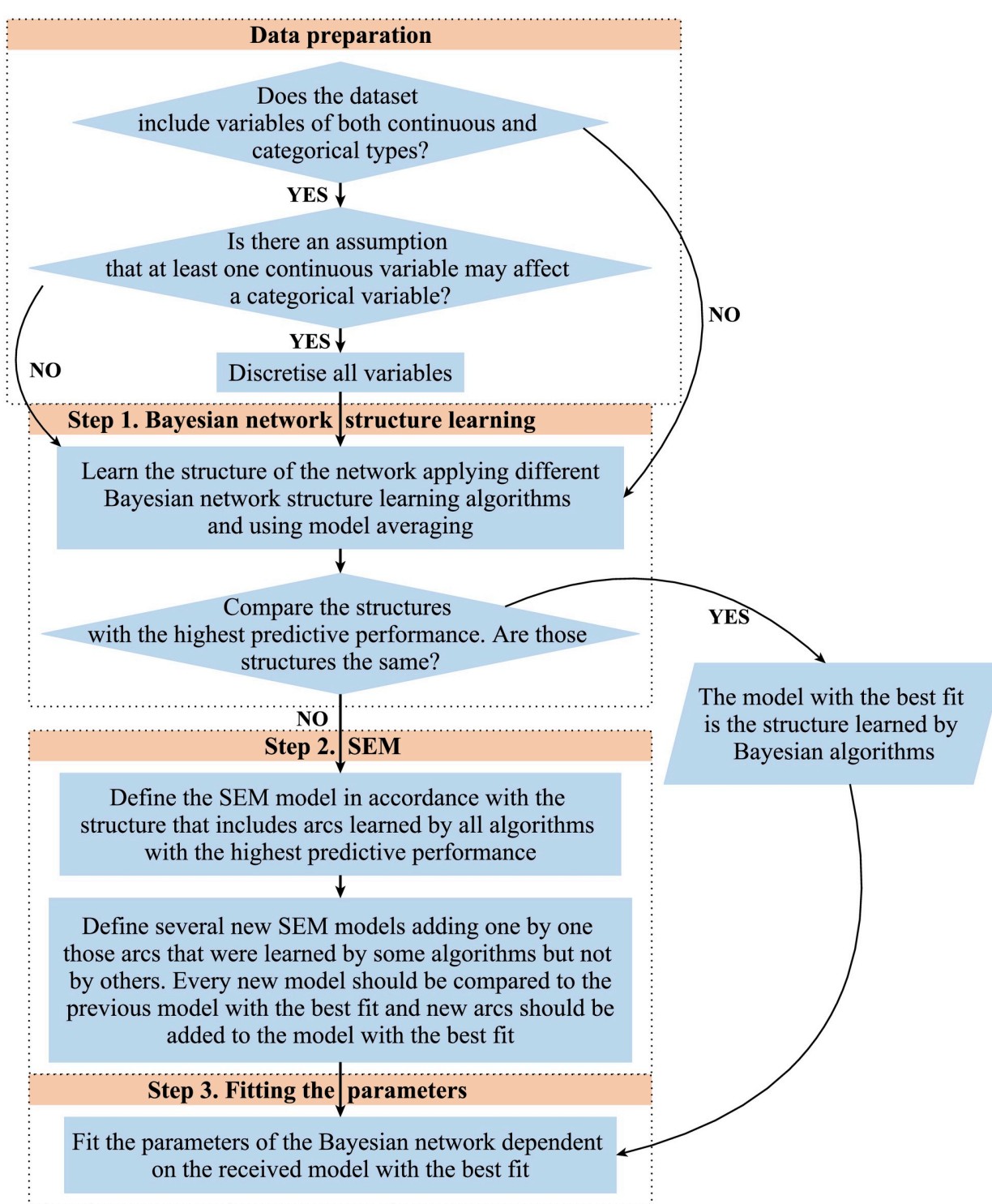

**Fig 1. The step-wise description of the procedure in analysing the data.**

learned partially directed structures (in comparison to the fully directed models found by the score-based and hybrid algorithms), only the results of the score-based and hybrid learning were used for further comparison.

The structures learned by score-based and hybrid algorithms were compared by evaluating the predictive performance of the models using cross-validation (see Table 3). In addition to that, the performance of the DAG structures received using the same algorithms but also, applying model averaging was assessed. Model averaging is used as a technique to acquire a model that gives better predictive performance and reduces over-fitting [47]. This method is often used when learning network structures on the datasets characterised by a limited number of observations and a high volume of noise [e.g., 59, 60]. Thus, the models were learned on the sets of 5000 network structures.

Comparing the predictive performance of all eight models (see Table 3), one can find TABU [56], HC [47] and H2PC [58] algorithms to outperform MMHC [57] on almost all dimensions of evaluation. This result is in line with the earlier observations of [49, 61]. In the

**Table 3. The results of cross-validation using different BN structures.**

| Algorithm type | Score-based | | Hybrid | | Score-based (averaged) | | Hybrid (averaged) | |
|---|---|---|---|---|---|---|---|---|
| Algorithm | TABU | HC | MMHC | H2PC | TABU | HC | MMHC | H2PC |
| Sensitivity | .844 | .843 | .842 | .835 | .845 | .844 | .840 | .844 |
| Specificity | .543 | .540 | .497 | .521 | .537 | .539 | .496 | .530 |
| Precision | .978 | .979 | .977 | .990 | .977 | .978 | .981 | .977 |
| Recall | .844 | .843 | .842 | .835 | .845 | .844 | .840 | .844 |
| F1 | .906 | .906 | .904 | .906 | .906 | .906 | .905 | .906 |
| Prevalence | .961 | .962 | .961 | .982 | .959 | .961 | .968 | .959 |
| Detection Rate | .811 | .811 | .809 | .820 | .810 | .811 | .813 | .810 |
| Detection Prevalence | .829 | .829 | .829 | .829 | .829 | .829 | .829 | .829 |
| Balanced Accuracy | .694 | .692 | .669 | .678 | .691 | .691 | .668 | .687 |
| BIC loss (SD) | 14.1388 (4e-04) | 14.139 (6e-04) | 14.2645 (8e-04) | 14.1529 (8e-04) | 14.1304 (6e-04) | 14.115 (7e-04) | 14.2859 (7e-04) | 14.1301 (7e-04) |
| BDE loss (SD) | 14.1386 (6e-04) | 14.1388 (5e-04) | 14.2646 (6e-04) | 14.1528 (8e-04) | 14.1313 (6e-04) | 14.1149 (5e-04) | 14.2858 (5e-04) | 14.1301 (7e-04) |
| Prediction error (SD) | 0.1681 (4e-04) | 0.1683 (4e-04) | 0.1718 (4e-04) | 0.1707 (3e-04) | 0.1682 (4e-04) | 0.1682 (3e-04) | 0.1711 (2e-04) | 0.1684 (4e-04) |
| Algorithm type | Score-based and hybrid (balanced) | | | | Averaged score-based and hybrid (balanced) | | | |
| Algorithm | TABU + H2PC | TABU + MMHC | HC + MMHC | HC + H2PC | TABU + H2PC | TABU + MMHC | HC + MMHC | HC + H2PC |
| Sensitivity | .835 | .838 | .838 | .835 | .845 | .829 | .829 | .843 |
| Specificity | .530 | .546 | .532 | .510 | .543 | .750 | 1.000 | .523 |
| Precision | .990 | .987 | .986 | .990 | .977 | 1.000 | 1.000 | .978 |
| Recall | .835 | .838 | .838 | .835 | .845 | .829 | .829 | .843 |
| F1 | .906 | .906 | .906 | .906 | .906 | .906 | .906 | .905 |
| Prevalence | .982 | .976 | .976 | .982 | .959 | 1.000 | 1.000 | .962 |
| Detection Rate | .820 | .818 | .817 | .820 | .810 | .829 | .829 | .811 |
| Detection Prevalence | .829 | .829 | .829 | .829 | .829 | .829 | .829 | .829 |
| Balanced Accuracy | .683 | .692 | .685 | .673 | .694 | .790 | .914 | .683 |
| BIC loss (SD) | 14.2078 (7e-04) | 14.1487 (8e-04) | 14.1488 (5e-04) | 14.2078 (5e-04) | 14.1899 (5e-04) | 14.2053 (7e-04) | 14.153 (8e-04) | 14.1391 (9e-04) |
| BDE loss (SD) | 14.2078 (5e-04) | 14.149 (7e-04) | 14.1489 (6e-04) | 14.2081 (5e-04) | 14.1899 (6e-04) | 14.2053 (7e-04) | 14.1537 (7e-04) | 14.1389 (8e-04) |
| Prediction error (SD) | 0.1711 (7e-04) | 0.1694 (4e-04) | 0.1693 (4e-04) | 0.1712 (5e-04) | 0.1683 (5e-04) | 0.1713 (1e-04) | 0.1712 (1e-04) | 0.1683 (5e-04) |

*Source*: ESS 2018 [35]. N = 27 379 individuals in 19 countries. The 10-fold cross-validation is applied to evaluate the predictive performance of the models. The top rows show the predictive performance of the structures learned by score-based and hybrid algorithms. The bottom rows show the predictive performance of the models received applying ensemble learning.

meanwhile, model averaging, indeed, shows to slightly increase the accuracy and recall, while reducing the prediction error in almost all of the cases.

In general, it is advised to consider the results of both score-based and hybrid algorithms when trying to learn the structures of networks consisting of categorical variables. While optimising the global model, score-based algorithms do not deal with local structure identification, which is considered when hybrid algorithms learn network structures. In that regard, it is worth recognising the results of the structure learning conducted by two types of algorithms, score-based and hybrid algorithms, as there is a higher risk of model over-fitting when using only score-based algorithms [49, p. 18]. Thus, the next step of the analysis was to apply structural equation modeling using the models found by the Bayesian structure learning algorithms to find significant paths. This was done with the goal to find the model that (1) provides high predictive performance and (2) reduced over-fitting and (3) that can serve as a theoretical foundation for future studies on online political participation predictors.

## Step 2: Structural equation modeling

The second step of the analysis was to test for the significance of the paths found by any pair of score-based—hybrid algorithms. For each pair of score-based—hybrid models, the initial structure of the network, which was fitted into the data by the means of structural equation modeling, included those arcs that appeared in both structures. After that, arcs present in one structure (e.g., a score-based model) but absent in another (e.g., a hybrid structure) were introduced in the model. As a result of computing the chi-square tests to compare the models [62], the structure with the best model fit was found and its performance was evaluated. Thus, Table 3 shows that the combination of the averaged TABU and H2PC structures, which was balanced by the means of structural equation modeling, gives the highest recall (i.e., 0.845) and lowest prediction error (0.1683) when compared with the performance of other balanced models. Moreover, this balanced model can be compared only with the averaged TABU structure when analysing the predictive performance. Thus, both models reach the recall of 0.845, while the averaged TABU structure has a slightly lower prediction error. Still, the balanced structure scores higher on the balanced accuracy, which is 0.694, compared to 0.691 for the averaged TABU structure. Thus, balancing the structures allows reaching goal (1), which was identified in the previous section, i.e., allows finding the model providing high predictive performance.

In order to test if balancing structures learned by score-based and hybrid algorithms decreases over-fitting, the method was used on the simulated dataset "A Logical Alarm Reduction Mechanism" (ALARM) [63] including 37 variables and 20 000 observations. Testing the method on the synthetic data allows finding the number of falsely identified arcs associated with each model. Thus, Fig 2 shows that in almost all of the cases, the accuracy of the models stays rather high, i.e., the number of correctly identified arcs stays with the range of 55%—65%. In the meanwhile, with the increase of the noise percentage in the data, the number of falsely identified arcs (i.e., false positives) rapidly grows. Moreover, as expected, score-based algorithms tend to over-fit the model.

While balancing score-based—hybrid pair of algorithms decreases accuracy only slightly, the number of falsely identified arcs drops significantly. In particular, Fig 2 shows that balancing the combination of the averaged TABU and H2PC structures allows reaching both, (1) high predictive performance and (2) reduced over-fitting, which supports the results reported in Table 3. In that regard, ensemble learning allows reaching both of the methodological goals identified earlier, as well as decreases the number of false positive arcs while keeping the number of correctly identified arcs high, thus, provides a good theoretical foundation for future studies on online political participation predictors.

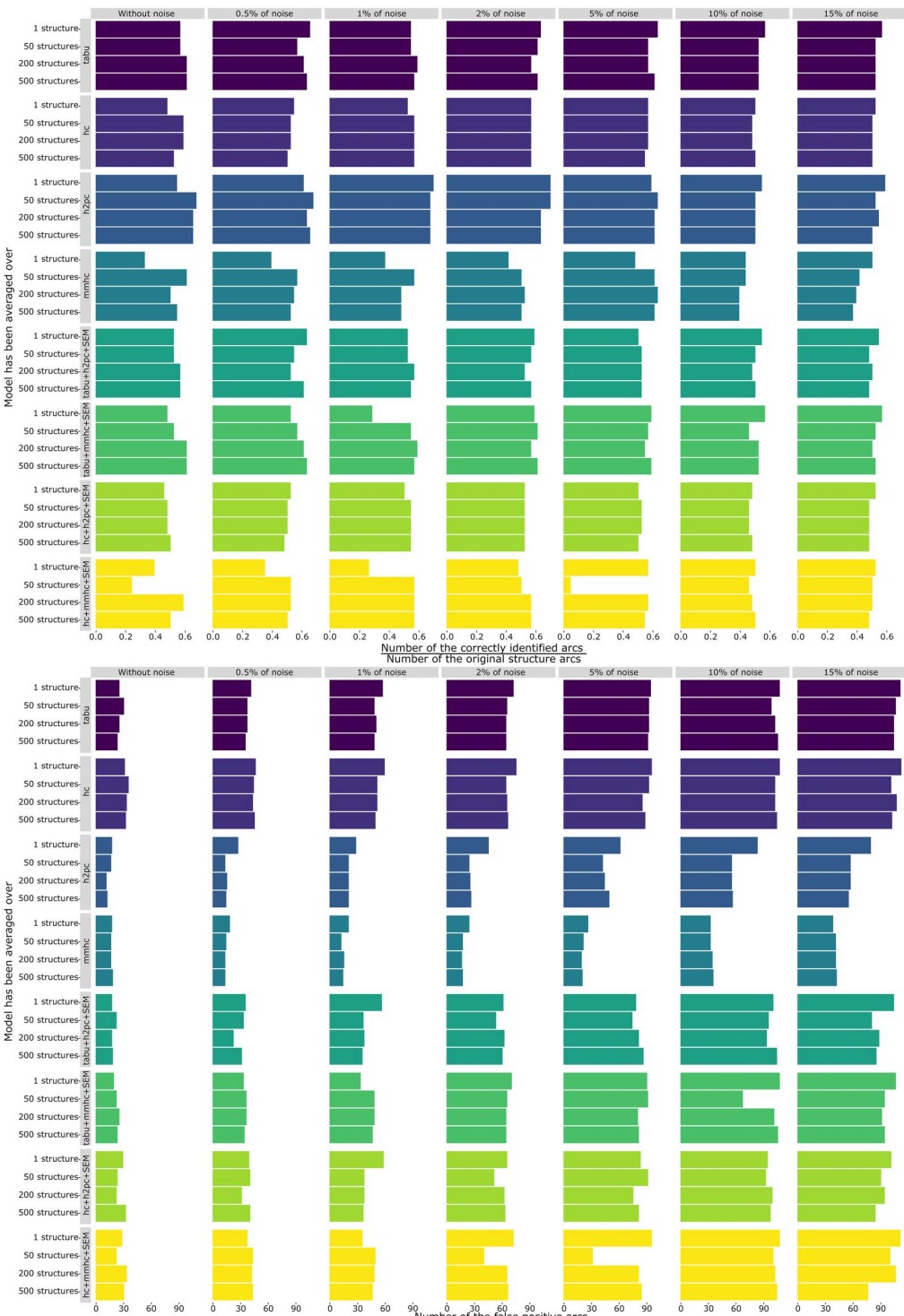

**Fig 2. Results of the over-fit testing on the synthetic dataset ALARM.** *Source*: [63]. N = 20 000 observations. *Notes*: The two-step approach was used to balance the structures. The over-fitting was tested on the original dataset and on the same dataset with an increased percentage of noise (i.e., 0.5%, 1%, 2%, 5%, 10% and 15% of noise). Eight rows on the top show the accuracy of the models in identifying the correct arcs. Thus, longer bars display higher accuracy. On the bottom, the rows show the number of falsely identified arcs. The shorter the bar the better as it signifies lower over-fitting.

Due to the fact that the balanced TABU + H2PC (both averaged on 5000 structures) model allowed reaching all of the identified methodological goals, this model was chosen for acquiring the set of conditional probability distribution tables.

## Step 3: Fitting the parameters

The third step of the analysis was to fit the parameters of the Bayesian network dependent on the received structure. In order to avoid receiving missing parameter estimates in the case when configurations of the discrete parents are not observed in the data [64], the Bayesian parameter estimation method was used to fit the parameters of the network. As a result, a set of probability distribution tables was acquired. Based on this data, it became possible to estimate the probability of a person to participate in online activism having some prior knowledge, e.g., one's level of political interest, political trust or placement of the left-right scale.

All statistical analyses were performed using the R 4.0.1 platform [65] and a number of additional packages, including bnlearn [64], lavaan [62], gRain [66] and psych [67]. The R scripts of the analysis and additional tests are provided as supplementary materials (i.e., S2 and S3 Files).

## Results

The results of the structure learning using the score-based Tabu and hybrid H2PC algorithms (averaged on 5000 structured) are illustrated by Fig 3.

Fig 3 shows that both algorithms agree on the relations between the majority of the nodes. Some arcs, however, are present in the structure learned by the Tabu algorithm, while being absent in the structure learned by H2PC (in Fig 3 they are orange). Moreover, two structures do not seem to agree on the directions of some arcs, i.e., arcs between political and social trust (thus, TABU algorithm found political trust to affect social trust, i.e., the relationship *political trust → social trust*, where *political trust* is a parent and *social trust* is its child; in the meanwhile, H2PC found inverse causality, i.e., the relationship *social trust → political trust*, where *social trust* is a parent and *political trust* is its child), internal and external political efficacy, internal political efficacy and education, as well as the relations between gender and internal political efficacy was determined only by the H2PC algorithm.

While conducting the robustness tests, i.e., learning the structures of the relations between the explanatory variables in regard to participation in signing petitions, contacting politicians and voting, other uncertainties in relation to the inverse or absent causality between participation in online activism and working in an NGO, absent causality between online participation and party identification, as well as in relation to a direct effect of internal political efficacy on party identification, arose. Those uncertainties were eliminated in the second step of the analysis by the means of structural equation modeling.

As a result of the structural equation modeling and comparison of the models based on the chi-square tests, the structure with the best model fit was distinguished. This structure is illustrated by Fig 4.

Based on the learned structure, participation in online activism is directly affected by only three explanatory variables, i.e., age, internal political efficacy and political interest, while political interest in itself is highly affected by internal political efficacy and age.

In accordance with the set of probability distribution tables, acquired as the result of fitting the parameters of the Bayesian network dependent on the received structure, the probability of participation in online activism changes depending on the age, internal political efficacy and political interest of a person (see Fig 5).

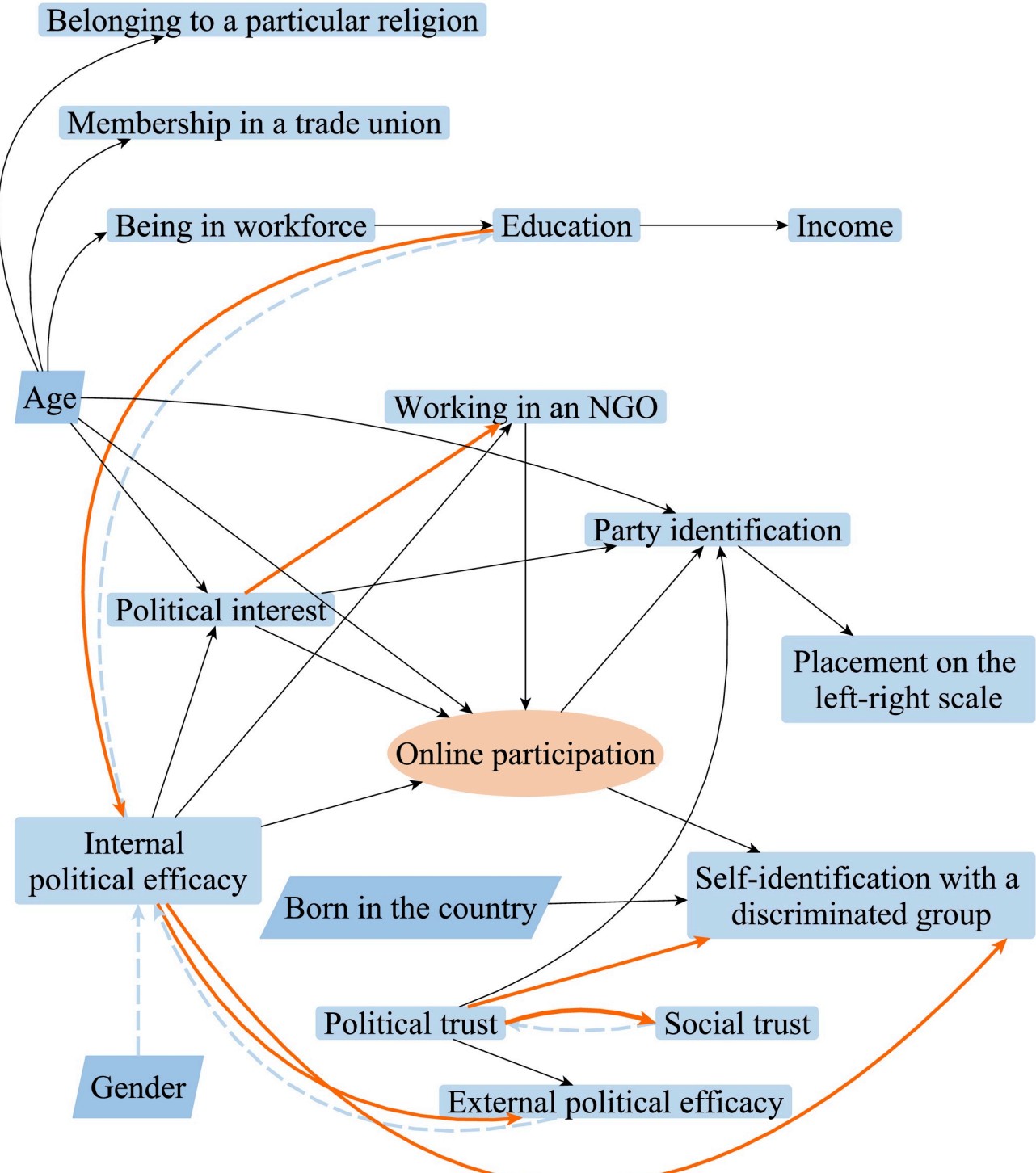

**Fig 3. Directed acyclic graphs of the relations between factors associated with participation in online activism.** *Source*: [35]. N = 27 379 individuals in 19 countries. *Notes*: Within Bayesian network analysis, score-based Tabu and hybrid H2PC algorithms were applied to analyse the data and learn the structure of the causal relations between the variables. Dashed blue lines represent false positives, i.e., edges that are not present in the structure learned by the Tabu algorithm but present in the structure learned by H2PC. Orange lines represent false negatives, i.e., edges that are present in the structure learned by the Tabu algorithm but absent in the structure learned by H2PC. The direction of each arc represents the orientation of the causality between two variables: e.g., in a relationship $A \rightarrow B$, $A$ is a parent and $B$ is its child. All the edges from the other nodes to "Age", "Gender" and "Born in the country" were blacklisted prior to learning the structure. No other edges were blacklisted. In the figure, those nodes that can only be parents have a darker blue color. The node "Country" (i.e., the country of the respondent's residency) is present in the structure but not depicted by the figure to facilitate the apprehension of the relations between the nodes of interest. All variables are individual-level variables.

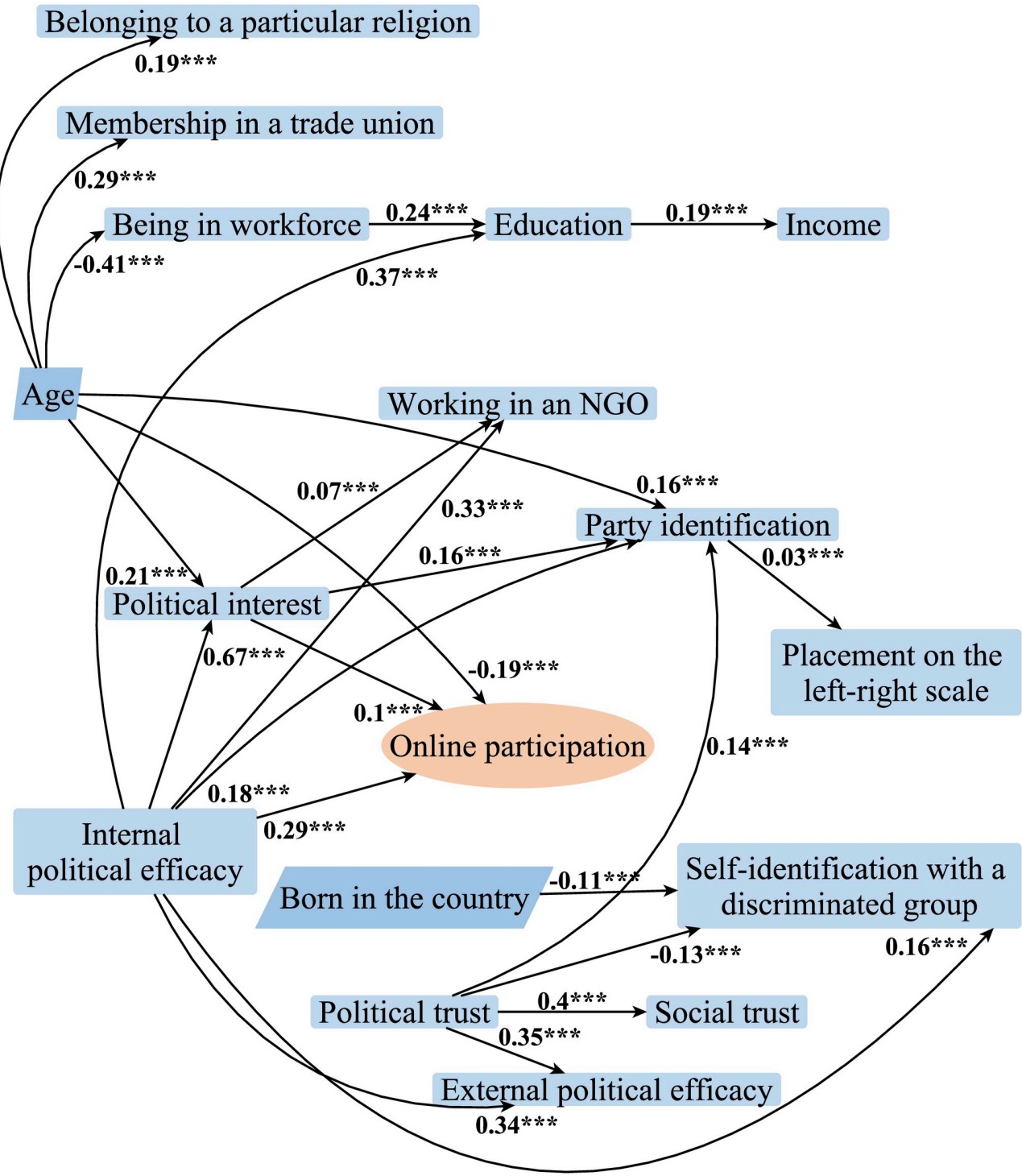

**Fig 4. Directed acyclic graph of the relations between factors associated with participation in online activism.** *Source*: ESS 2018 [35]. N = 27 379 individuals in 19 countries. *Notes*: Structural equation modeling was applied to analyse the data. Entities depicted in association with the edges are parameter estimates of the structural equation modeling. Sign.: *p < 0.05; **p < 0.01; ***p < 0.001. All variables are individual level variables.

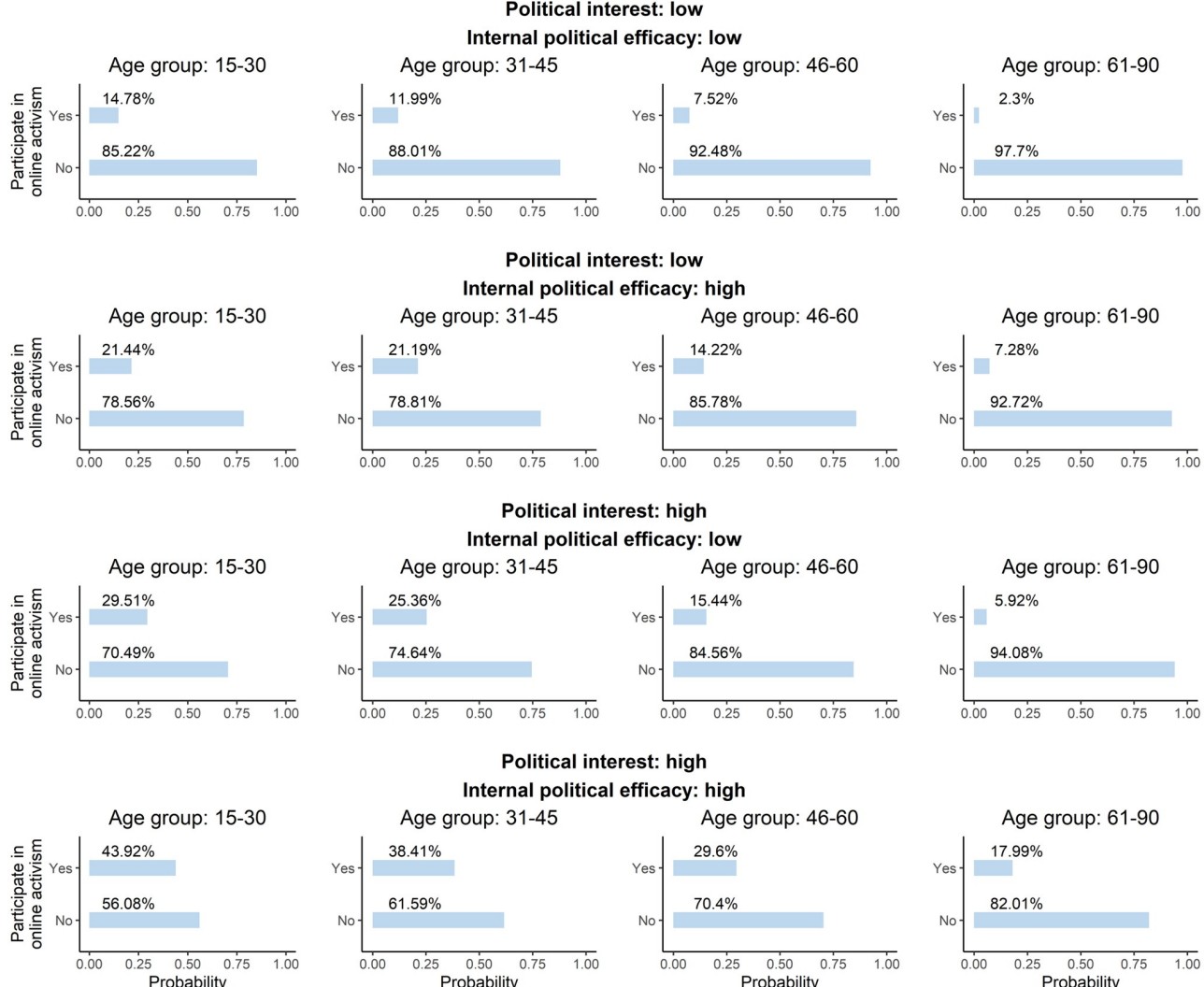

**Fig 5. Probability distribution table of participation in online activism.** *Source*: ESS 2018 [35]. N = 27 379 individuals in 19 countries. *Notes*: Bayesian parameter estimation, conditional on the acquired structure of the network, was applied to analyse the data. Entities are the probability of participation in online activism in percentage.

Fig 5 shows that the group of people that is more likely to participate in online activism are those between 15 and 30 years old with high political interest and internal political efficacy (about 44 out 100 people of that group are likely to participate in online activism). In the meanwhile, the probability of participation online in the age group least expected to participate, i.e., between 61 and 90 years old, grows from 2,3% to 17,99% with the increase in political interest and internal political efficacy. That suggests significant growth in the probability of online political participation dependent on the political interest and internal political efficacy that is partly in line with the previous suggestions of [12].

Fig 5 also suggests that political interest has a higher influence on online political participation than internal political efficacy in all age groups except for those between 61 and 90 years old. For the elderly, internal political efficacy, which is party operationalises access to resources, has a bigger influence on participation in online activism (the probability of online participation in the group of people with high internal political efficacy is 7,28%) than political

interest (the probability of online participation in the group of people with high political interest is 5,92%).

The acquired structure of the network (see Fig 4) suggests that the rest of the variables are independent of participation in online activism given age, internal political efficacy and political interest.

The causal relations between the explanatory variables seem rather interesting regarding the results of earlier studies. Thus, for instance, nodes that are often referred to as recruitment variables [12], i.e. membership in a trade union, belonging to a particular religion, being in workforce and working in a non-governmental organisation, in practice, indicate the age of a person rather than affect online political participation. Moreover, only one of those variables, working in an NGO, depends on political motivation variables, i.e. political interest and political efficacy, and within robustness tests, showed to affect at most participation in contacting politicians while also was found to be dependent on participation in signing petitions.

Party identification, placement on the left-right scale, self-identification with a discriminated group, political and social trust and external political efficacy also appear to be independent of participation in online activism given internal political efficacy and age.

Despite the fact that the majority of the variables associated with participation in online activism does not have a direct influence on the response variable, having prior knowledge of some of those characteristics can suggest a higher or lower probability of a person to participate in online activism. Thus, without having any prior knowledge about a person, it is expected that there is a 17,03% chance that the person participates online. Fig 6 shows the probability distributions of all the network variables if there is no prior knowledge about a person.

If there is some prior knowledge about a person, e.g., it is a person with the low income, graduate level of education, working in a non-governmental organisation, placing oneself on the left of the left-right scale and having a low political trust, the probability of online participation is expected to increase from 17,03% to 24,48%. Thus, even without having any prior knowledge about those factors that directly influence participation in online activism, i.e., age, internal political efficacy and political interest, the probability of participation is expected to be higher. In that case, the probabilities of other events would also change. Thus, the probability of having high internal political efficacy would grow from 45,67% to 80,64%, the probability of having a high political interest would increase from 49,02% to 74,99%, the probability of self-identification with a discriminated group would change from 7,30% to 12,41%, the probability of having high social trust would drop from 58,14% to 44,86% (see S6 Fig to know how the probability distributions of other variables would change).

## Conclusion

While conducting the analysis of the survey data, it became evident that using Bayesian network analysis as the only method of research can produce unreliable results. This paper presents a three-step approach to acquire a reliable structure of causal relations between characteristics operationalised by survey questionnaires.

The analysis showed that while agreeing on the majority of the causal relations of networks based on survey data, constrained-based, score-based and hybrid algorithms used for structure learning still do not agree on some relations. That being the case, it is necessary to refine the results of the Bayesian network structure learning comparing received structures by the means of structural equation modeling.

In this analysis, the accuracy of the learned structure was also constrained by the imbalanced data. Thus, out of 27,379 observations, only 4,687 individuals (17,12%) participate in

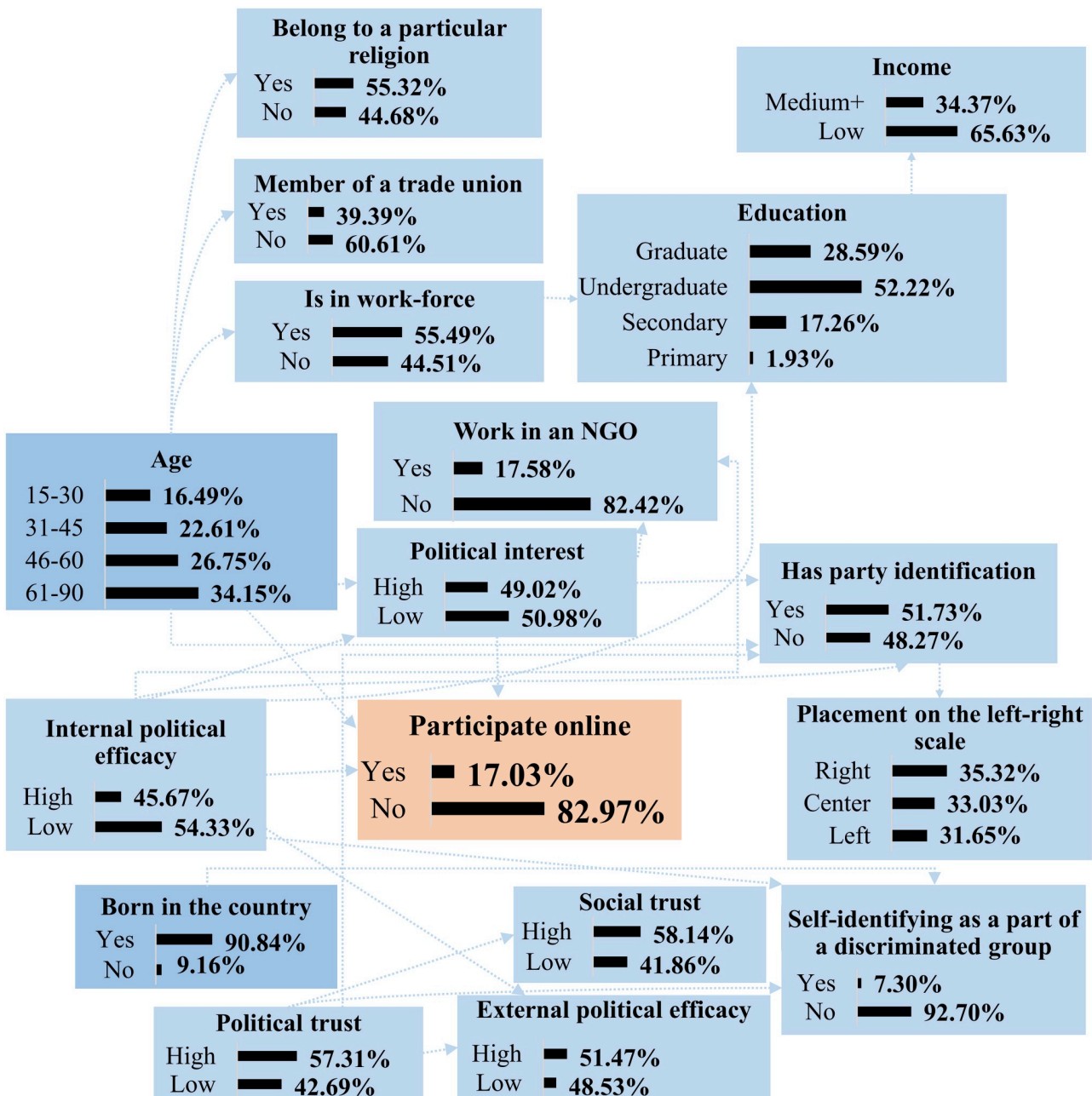

**Fig 6. Probability distribution of all factors associated with participation in online activism.** *Source*: ESS 2018 [35]. N = 27 379 individuals in 19 countries. *Notes*: Bayesian parameter estimation, conditional on the acquired structure of the network, was applied to analyse the data. Entities are the probabilities of events in percentage.

online activism. Hence, it became also necessary to conduct robustness tests adding participation in signing petitions, contacting politicians and voting as other outcome variables. That allowed to significantly increase the number of observations associated with political participation (e.g., out of 27,323 observations, 9,426 people (34,5%) participate in signing petitions and online activism) and redefine the causal relation between some of the nodes.

Applying a three-step approach and conducting robustness and validity tests to analyse the survey data, it became possible to receive a reliable structure of the causal relations between

the variables associated with participation in online activism. The acquired structure (see Fig 4) suggests causality dissimilar to that reported before. Hence, despite the fact that the structure partly supports the Civic Voluntarism Model (CVM) developed by [12] in relation to the effects of internal political efficacy and political interest, the effect of political interest is still mediated by the indirect effects of internal political efficacy and age via political interest. Furthermore, in regard to other factors, the causal relation is absent. For instance, recruitment does not seem to increase participation in online activism. When performing the robustness tests, it became evident that similar causal relations are also in place in regard to participation in signing petitions, contacting politicians and voting. Moreover, robustness tests showed that those who sign petitions also get mobilised into other activities, i.e., start working in an NGO, which is a reverse causality than the one expected by [12].

Political trust does seem to be independent of participation in online activism contrary to some of the previous suggestions [14, 38–41]. Furthermore, within the structure, the variable is only affected by the country of residence and independent of political interest in the European context.

Fig 4 also shows that such resources as education and income do not contribute to increasing the probability of participation in online activism. However, due to the fact that internal political efficacy, which in this analysis, operationalises access to resources and acquired skills that allow a person to use such resources, has a direct effect on online participation, it is suggested that access to resources other than money or education increases the probability of online activism participation in Europe. That result is also partly in line with the CVM [12].

Despite the fact that Bayesian network analysis allowed to distinguish the structure of causal relations between the variables associated with participation in online activism and determine which variables directly affect participation, 44% is the highest probability of a person to participate in online activism given prior knowledge of the variables examined by political participation scholars (see Fig 5). Such a result may propose that rather than characteristics, personal motives [68] and emotions [69], factors highlighted by the social movement literature, or other factors, which are not yet reported, stimulate political participation. In that regard, there is much work to be done in order to distinguish the factors that directly affect political participation and allow predicting the last one. It seems necessary to examine such factors outside the European context as well since the present study showed that access to resources plays a big role in stimulating political participation as suggested by previous research [12, 68]. This paper showed how causal relations can be inferred using Bayesian network analysis in combination with structural equation modeling. Other methods can also be considered when stabilising the results of Bayesian network structure learning.

## Supporting information

**S1 Fig. Directed acyclic graphs of the relationships between factors associated with participation in petition signing.** *Source*: [35]. *N* = 27 366 individuals in 19 countries. *Notes*: Within Bayesian network analysis, score-based Tabu and hybrid H2PC algorithms were applied to analyze the data and learn the structure of the causal relationships between the variables. Dashed blue lines represent false positives, i.e., edges that are not present in the structure learned by the Tabu algorithm but present in the structure learned by H2PC. Orange lines represent false negatives, i.e., edges that are present in the structure learned by the Tabu algorithm but absent in the structure learned by H2PC. All the edges from the other nodes to "Age", "Gender" and "Born in the country" are blacklisted prior to learning the structure. In the figure, those nodes that can only be parents have a darker blue color. The node "Country" (i.e., the country of the respondent's residency) is present in the structure but not depicted by the

figure to facilitate the apprehension of the relationships between the nodes of interest. All variables are individual-level variables.
(TIF)

**S2 Fig. Directed acyclic graphs of the relationships between factors associated with participation in online activism and petition signing.** *Source*: [35]. N = 27 323 individuals in 19 countries. *Notes*: Within Bayesian network analysis, score-based Tabu and hybrid H2PC algorithms were applied to analyze the data and learn the structure of the causal relationships between the variables. Dashed blue lines represent false positives, i.e., edges that are not present in the structure learned by the Tabu algorithm but present in the structure learned by H2PC. Orange lines represent false negatives, i.e., edges that are present in the structure learned by the Tabu algorithm but absent in the structure learned by H2PC. All the edges from the other nodes to "Age", "Gender" and "Born in the country" are blacklisted prior to learning the structure. In the figure, those nodes that can only be parents have a darker blue color. The node "Country" (i.e., the country of the respondent's residency) is present in the structure but not depicted by the figure to facilitate the apprehension of the relationships between the nodes of interest. All variables are individual-level variables.
(TIF)

**S3 Fig. Directed acyclic graphs of the relationships between factors associated with participation in contacting politicians.** *Source*: [35]. N = 27 397 individuals in 19 countries. *Notes*: Within Bayesian network analysis, score-based Tabu and hybrid H2PC algorithms were applied to analyze the data and learn the structure of the causal relationships between the variables. Dashed blue lines represent false positives, i.e., edges that are not present in the structure learned by the Tabu algorithm but present in the structure learned by H2PC. Orange lines represent false negatives, i.e., edges that are present in the structure learned by the Tabu algorithm but absent in the structure learned by H2PC. All the edges from the other nodes to "Age", "Gender" and "Born in the country" are blacklisted prior to learning the structure. In the figure, those nodes that can only be parents have a darker blue color. The node "Country" (i.e., the country of the respondent's residency) is present in the structure but not depicted by the figure to facilitate the apprehension of the relationships between the nodes of interest. All variables are individual-level variables.
(TIF)

**S4 Fig. Directed acyclic graphs of the relationships between factors associated with participation in voting.** *Source*: [35]. N = 25 404 individuals in 19 countries. *Notes*: Within Bayesian network analysis, score-based Tabu and hybrid H2PC algorithms were applied to analyze the data and learn the structure of the causal relationships between the variables. Dashed blue lines represent false positives, i.e., edges that are not present in the structure learned by the Tabu algorithm but present in the structure learned by H2PC. Orange lines represent false negatives, i.e., edges that are present in the structure learned by the Tabu algorithm but absent in the structure learned by H2PC. All the edges from the other nodes to "Age", "Gender" and "Born in the country" are blacklisted prior to learning the structure. In the figure, those nodes that can only be parents have a darker blue color. The node "Country" (i.e., the country of the respondent's residency) is present in the structure but not depicted by the figure to facilitate the apprehension of the relationships between the nodes of interest. All variables are individual-level variables.
(TIF)

**S5 Fig. Directed acyclic graph of the relationships between factors associated with participation in online activism.** *Source*: [35]. N = 27 379 individuals in 19 countries. *Notes*:

Structural equation modeling was applied to analyze the data. Only those arcs that were determined by both Tabu and H2PC algorithms are present in the model. Entities depicted in association with the edges are parameter estimates of the structural equation modeling. Sign.: $^*p < 0.05$; $^{**}p < 0.01$; $^{***}p < 0.001$. All variables are individual level variables.
(TIF)

**S6 Fig. Probability distribution of all factors associated with participation in online activism.** *Source*: ESS 2018 [35]. N = 27 379 individuals in 19 countries. *Notes*: Bayesian parameter estimation, conditional on the acquired structure of the network, was applied to analyse the data. Entities are the probabilities of events in percentage. The following conditional probability query was applied: education is "graduate", placement on the left-right scale is "left", work in an NGO is "yes", political trust is "low" and income is "low".
(TIF)

**S7 Fig. Probability distribution of all factors associated with participation in online activism.** *Source*: [35]. N = 27 379 individuals in 19 countries. *Notes*: Bayesian parameter estimation, conditional on the acquired structure of the network, was applied to analyze the data. Entities are the probabilities of events in percentage. The following conditional probability query was applied: age is "31–45", political interest is "high", political trust is "high", social trust is "high", internal political efficacy is "high" and born in the country of residence is "yes".
(TIF)

**S1 File. Supplementary information.** The document provides additional information on the data handling, methods and results and complements the main text of the manuscript.
(PDF)

**S2 File. Supplementary R script.** The R script used for the analysis.
(PDF)

**S3 File. Supplementary R script.** The R script used to compare the predictive performance of the models and to test a two-fold approach in network structure learning, i.e., the combination of Bayesian structure learning and structural equation modeling, on simulated data.
(PDF)

## Author Contributions

**Conceptualization:** Elizaveta Kopacheva.

**Formal analysis:** Elizaveta Kopacheva.

**Methodology:** Elizaveta Kopacheva.

**Validation:** Elizaveta Kopacheva.

**Visualization:** Elizaveta Kopacheva.

**Writing – original draft:** Elizaveta Kopacheva.

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
