## [Decision Letter · Decision Letter 0]

1 Oct 2021

PONE-D-21-05594

Predicting online participation through Bayesian network analysis

PLOS ONE

Dear Dr. Kopacheva,

Thank you for submitting your manuscript to PLOS ONE. After careful consideration, we feel that it has merit but does not fully meet PLOS ONE’s publication criteria as it currently stands. Therefore, we invite you to submit a revised version of the manuscript that addresses the points raised during the review process.

Both reviewers have made comments on a number of issues requiring your attention. Specifically:

- Clarification of methodological aspects (most comments by both reviewers): More details should be provided about the applied methodology. In this sense, pleasenotice PLOS ONE's publication criterion #3 ().

- Results and their interpretation: Clarify the issues raised by Reviewer 1 in his comments #8, 9 and 10.

Additionally, consider the request for more information on Bayesian networks made by Reviewer 2 in his/her comment #2.

We look forward to receiving your revised manuscript.

Kind regards,

Sergi Lozano

Academic Editor

PLOS ONE

Journal Requirements:

2. Thank you for stating the following financial disclosure: "NO"

3. Thank you for stating the following in your Competing Interests section: "NO"

4. Please upload a copy of Supporting Information Table 1 and 2 which you refer to in your text on page 9.

Reviewers' comments:

Reviewer's Responses to Questions

**Comments to the Author**

1. Is the manuscript technically sound, and do the data support the conclusions?

Reviewer #1: Yes

Reviewer #2: Yes

2. Has the statistical analysis been performed appropriately and rigorously? 

Reviewer #1: Yes

Reviewer #2: Yes

3. Have the authors made all data underlying the findings in their manuscript fully available?

Reviewer #1: Yes

Reviewer #2: Yes

4. Is the manuscript presented in an intelligible fashion and written in standard English?

Reviewer #1: Yes

Reviewer #2: Yes

5. Review Comments to the Author

Reviewer #1: 1.- The paper reports the application of Bayesian networks to find causal relationships between variables affecting online political participation. The authors propose a three-step methodology to find these causal relationships: a. finding the network structure, b. finding the statistical significance of those relationships using structural equation modeling, and c. fitting the distributions or parameters.

2. I think the study represents an interesting methodological application for modeling the participation of individuals in political issues. As the authors point out, the use of Bayesian networks is limited to the availability of large amounts of data. Consequently, it is not often practical in the social science data domain. Nevertheless, the ESS provides an excellent opportunity to test this type of tool. The study stands out for its originality. I can currently point out a few improvements that the authors could consider and clarify some doubts regarding the methodology.

Comments:

3.- As the authors rightly point out, the structure of a Bayesian network can be affected by imprecise or missing observations.

In this regard, in the Results section, I found no information regarding missing data. How much did missing data were there?, on which variables?

What was done with the observations in cases where there was a missing measurement or missing variable?

Regarding measurement imprecision, it would be useful to clarify at least for the most important variables (online participation, political participation, political participation, etc.) how these variables were measured in the ESS.

4.- At the structure learning stage:

A typical issue that needs to be dealt with is selecting the best structure for the Bayesian network.

In this process, as mentioned by the authors, it is possible to use several algorithms. They use score-based and hybrid algorithms. Is it possible to report a table with more accurate information or some statistics of the goodness-of-fitness of the algorithms to compare the performance?

On the other hand, as a means of robustness, I understand, they use several learned structures to predict the variable "online participation." The idea is to evaluate these models to get the best of them.

Based on what performance measure do the authors evaluate the predictive ability?

In summary, it would be nice to explain how they found the best set of network structures in more detail.

5.- how do you prevent the problem of overfitting? do you use any cross-validation scheme?

6.- According to this, step 2 (SEM) identifies those edges or arcs of the Bayesian network that are statistically significant.

If we assume that we are only interested in the model that performs best in predicting the dependent variable, then what do we gain from step 2?

I assume that the authors are not only interested in predictive ability, but also in "explaining" the online participation variability between observations.

7.- From the Bayesian network, we obtain causality information. In SEM we see if there is statistical significance in the network arcs. So how do we interpret causality between two variables that are not significant? How do we interpret a pair of variables whose relationship is significant but without connecting arcs in the network?

8.- The results section reports the results of the directed acyclic graphs. The section reports the Edges present in the structures and that are influential in online activism. It also reports the inference parameters (conditional probability distributions). However, I do not find results concerning "prediction ." In this regard, I would like to differentiate between the knowledge we gain in interpreting the network structure and the ability of this model to predict.

What happens with the model's performance (Bayesian network learned) to predict the online activism variable in a different sample than the one used for the learning process?

My understanding is that the suggested three-step approach allows adding robustness and validity to the results, especially for survey data. However, I wonder how much the predictive capacity increases between a model found with this methodology and another without this methodology. This is of interest because the title of this paper is precisely related to the "prediction" of online participation.

9.- In the results section on page 8, Lines 289-301, the authors report sensitivity. They indicate how much online participation could increase or decrease when the level of other variables changes. I believe that much could be gained from the results by specifying a more detailed sensitivity analysis that would allow us to understand the most critical factors on online participation on the dependent variable in terms of intensity and direction.

Minor comments:

10.- This seems obvious, but for a reader not so knowledgeable in Bayesian networks, it might be nice to interpret the directionality (causality) of the Bayesian network edges.

Reviewer #2: This article scrutinizes the causal relations between the variables associated with participation in online activism and introduces a three-step approach in learning a reliable structure of the participation preconditions’ network to predict political participation. The authors use Bayesian network analysis and structural equation modeling to stabilize the structure of the causal relations.

Please consider the following points for revision:

1. clarify which algorithm was used for the discretization

2. Step 1 should include more information on Bayesian networks. mathematical model, different networks and a mathematical description of the algorithms used.

https://www.sciencedirect.com/science/article/pii/S0167739X19303322

https://link.springer.com/article/10.1023/A:1007465528199

3. I do not understand how the authors choose the high scores of the networks. what is this? what is the class label used in the Bayesian network?

4. Authors should include more details of the data used, especially when it is used in the Bayesian network, it is not clear.

6. PLOS authors have the option to publish the peer review history of their article (what does this mean?). If published, this will include your full peer review and any attached files.

Reviewer #1: **Yes: **Mauricio A. Valle

Reviewer #2: No

---

## [Author Response · Author response to Decision Letter 0]

5 Nov 2021

I thank the reviewers and the associate editor for their constructive comments. I have addressed all of them and modified the paper accordingly. Please, find the modified manuscript (i.e., Revised Manuscript with Track Changes. pdf) and the detailed answer to the questions in the file Response to reviewers.pdf. Thank you!

---

## [Decision Letter · Decision Letter 1]

25 Nov 2021

PONE-D-21-05594R1Predicting online participation through Bayesian network analysisPLOS ONE

Dear Dr. Kopacheva,

Thank you for submitting your manuscript to PLOS ONE. As you can see below, the reviewers' are satisfied with your revision of the manuscript. Still, Reviewer 2 has pointed out to two (very specific) issues. According to his/her comments, please check Fig. 2 and consider moving Table 3 to the Supplementary (as it does not contain specific information needed to follow the main text). We invite you to submit a revised version of the manuscript that addresses the points raised during the review process.

We look forward to receiving your revised manuscript.

Kind regards,

Sergi Lozano

Academic Editor

PLOS ONE

Journal Requirements:

Reviewers' comments:

Reviewer's Responses to Questions

**Comments to the Author**

1. If the authors have adequately addressed your comments raised in a previous round of review and you feel that this manuscript is now acceptable for publication, you may indicate that here to bypass the “Comments to the Author” section, enter your conflict of interest statement in the “Confidential to Editor” section, and submit your "Accept" recommendation.

Reviewer #1: All comments have been addressed

Reviewer #2: All comments have been addressed

2. Is the manuscript technically sound, and do the data support the conclusions?

Reviewer #1: Yes

Reviewer #2: Yes

3. Has the statistical analysis been performed appropriately and rigorously? 

Reviewer #1: Yes

Reviewer #2: Yes

4. Have the authors made all data underlying the findings in their manuscript fully available?

Reviewer #1: Yes

Reviewer #2: Yes

5. Is the manuscript presented in an intelligible fashion and written in standard English?

Reviewer #1: Yes

Reviewer #2: Yes

6. Review Comments to the Author

Reviewer #1: (No Response)

Reviewer #2: The manuscript has improved. I have some minor comments:

1. Figure 2 is illegible. Please improve the quality.

2. Table 3 is very long. Please, move table 3 to supplementary material.

7. PLOS authors have the option to publish the peer review history of their article (what does this mean?). If published, this will include your full peer review and any attached files.

Reviewer #1: **Yes: **Mauricio A. Valle

Reviewer #2: No

---

## [Author Response · Author response to Decision Letter 1]

3 Dec 2021

I thank the reviewers and the associate editor for their comments. In particular, Reviewer 2 mentioned two specific issues: 

1. That ``Figure 2 is illegible": I have addressed this issue and improved the quality of the figure;

2. That ``Table 3 is very long": I have moved Table 3 to the supplementary material file (S1 File). 

Thank you!

---

## [Editor Report · Decision Letter 2]

9 Dec 2021

Predicting online participation through Bayesian network analysis

PONE-D-21-05594R2

Dear Dr. Kopacheva,

We’re pleased to inform you that your manuscript has been judged scientifically suitable for publication and will be formally accepted for publication once it meets all outstanding technical requirements.

Kind regards,

Sergi Lozano

Academic Editor

PLOS ONE
---

## [Editor Report · Acceptance letter]

14 Dec 2021

PONE-D-21-05594R2 

Predicting online participation through Bayesian network analysis 

Dear Dr. Kopacheva:

I'm pleased to inform you that your manuscript has been deemed suitable for publication in PLOS ONE. Congratulations! Your manuscript is now with our production department. 

Kind regards, 

on behalf of

Dr. Sergi Lozano 

Academic Editor

PLOS ONE